# OpenReview forum: "Goal-Aware Identification and Rectification of Misinformation in Multi-Agent Systems"
_ICLR.cc/2026/Conference — ICLR 2026 Poster_

### Official Review · Reviewer_9fCa · 2025-10-16

**Soundness:** 2
**Presentation:** 2
**Contribution:** 2
**Rating:** 4
**Confidence:** 4

**Summary:**

This paper makes two primary contributions. First, they introduce MISINFOTASK, a new benchmark dataset comprising 108 tasks designed specifically to evaluate MAS robustness against covert misinformation. The dataset includes plausible but fallacious arguments for each task, along with ground truth information. Second, they propose ARGUS, a training-free, two-stage defense framework.

**Strengths:**

1.  This paper evaluates the robustness of MAS against misinformation using  Tool Injection, Prompt Injection and RAG Injection.

2.  This paper builds a corrective agent to guard the misinformation in MAS.

**Weaknesses:**

1. The initial localization phase relies exclusively on Edge Betweenness Centrality, a purely topological metric, to identify critical channels. This strategy assumes that misinformation will travel along the most central paths. However, a sophisticated adversary could easily evade this initial detection by injecting misinformation through less central, seemingly unimportant edges, allowing it to propagate for at least one full round before content-aware mechanisms are activated.

2. The effectiveness of the adaptive re-localization hinges entirely on the corrective agent's ability to accurately infer the misinformation's intent-driven goal $g_{mis}$. If this inference is flawed or inaccurate, the subsequent calculation of information relevance $Score_{rel}$ will be based on an incorrect premise.

3. The entire defense rests on the strong assumption that the corrective agent's internal, parameterized knowledge is factually superior to the incoming information.

4. The comprehensive score for channel importance is calculated using fixed weights (α=0.2, β=0.2, γ=0.6). These static values may not be optimal across different MAS topologies, task types, or evolving adversarial strategies.

5. The paper states that the dataset underwent a rigorous manual review process after AI generation. However, the methodology lacks transparency. Crucial details are omitted, such as the qualifications of the human experts, the number of reviewers per entry, the specific guidelines for filtering content, and any inter-annotator agreement metrics. This makes it difficult to independently assess the quality, consistency, and objectivity of the final dataset.

6. The dataset's size of 108 entries is relatively small for a benchmark intended to test broad generalization. Furthermore, since the initial data was generated by a single LLM (GPT-4o), there is a risk that the misinformation is inadvertently tailored to the specific failure modes of that model family.

7. The dataset generation prompt explicitly targets misinformation that contradicts well-established facts likely learned during pre-training. This approach overlooks the challenge of dynamic misinformation related to recent events, evolving topics, or time-sensitive data that falls outside an LLM's static knowledge base. This limits the dataset's applicability to real-world scenarios where misinformation is often timely and ephemeral.

**Questions:**

1. The number of monitored channels was set to k=M-1, meaning nearly every channel was monitored. This implies significant computational and financial costs. Have you considered or experimented with more efficient implementations, such as a sampling strategy for monitoring or a lightweight initial check to triage messages before triggering the full, resource-intensive CoT analysis?

2. The corrective agent's rectification strategy relies on a CoT prompting method guided by heuristic principles like "root cause analysis" and "cognitive reframing". Could you discuss the generalizability of this prompting strategy?

---

> ### Author Response · Authors · 2025-11-21
> **Response to Reviewer 9fCa (1/2)**
>
> Dear **Reviewer 9fCa**:
>
> We sincerely thank you for your thoughtful feedback and valuable suggestions on our work!
>
> We are pleased to clarify and answer the questions and provide some additional experimental results.
>
> # For Weakness 1
>
> Thank you for your question. We acknowledge that the scenario is indeed possible.
>
> However, in turn, misinformation propagating along less central pathways also has a limited immediate impact on the overall MAS. This reduced centrality makes the misinformation easier to detect and rectify during subsequent, content-aware operational rounds. We maintain that relying on a purely topological metric for the defense's initial startup phase is the optimal trade-off, as it maximizes the coverage of the most critical information pathways.
>
> # For Weakness 2
>
> Thank you for your comment. We agree that errors in goal inference may lead to miscalculation of information relevance.
>
> However, this is precisely why our localization relies on a weighted combination of three metrics. If the information relevance score is flawed, the other two metrics (topological importance and usage frequency) still influence the final localization decision, preventing the process from being entirely limited by a potentially incorrect goal inference.
>
> Furthermore, the experiments presented in Figure 4 demonstrate the final goal inference capability of ARGUS, confirming that our module maintains high accuracy even in covert misinformation scenarios.
>
> # For Weakness 3
>
> Thank you for your question.
>
> First, we consider this assumption reasonable because, as explicitly defined in Section 2.3, we frame misinformation as content that contradicts the factual knowledge implicitly stored within the LLM's parameters.
>
> However, we disagree that the defense entirely rests on this single assumption. The rectification process also includes several independent checks beyond factual comparison: identifying potential logical inconsistencies, checking for deviations from common sense, scrutinizing ambiguous phrasing, and executing goal-aware intent inference. Newer or more complex misinformation scenarios can be detected through these compensatory heuristics.
>
> Furthermore, ARGUS is designed to be extensible: dynamic knowledge can be seamlessly introduced to our MAS and corrective agent via RAG integration, addressing the implicit concern regarding time-sensitive facts.
>
> # For Weakness 4
>
> We agree that static hyperparameters may not achieve universal optimality across diverse MAS topologies, task types, and adversarial strategies. We established the ratio of $\alpha=0.2, \beta=0.2, \gamma=0.6$ to determine a universally effective setting based on the logical contribution of each score:
>
> - The **goal-aware misinformation relevance score** ($\gamma=0.6$) receives the highest weight as it provides the most direct evidence for locating active attack channels.
> - The topological importance ($\alpha=0.2$) and channel frequency scores ($\beta=0.2$) are assigned lower, equal weights, as they represent less dynamic or direct evidence.
>
> To empirically validate this rationale, we performed targeted ablation experiments by sequentially eliminating one or two importance scores, observing the resulting performance degradation in ARGUS.
>
> |                       | MT   | TSR   |
> | --------------------- | ---- | ----- |
> | ARGUS                 | 3.73 | 75.86 |
> | w/o $\alpha$            | 4.14 | 70.37 |
> | w/o $\beta$             | 3.76 | 72.22 |
> | w/o $\gamma$            | 4.59 | 68.52 |
> | w/o $\beta$ and $\gamma$  | 4.34 | 69.44 |
> | w/o $\alpha$ and $\gamma$ | 4.79 | 67.59 |
> | w/o $\alpha$ and $\beta$  | 3.91 | 73.14 |
>
> **This experiment and the corresponding analysis have been integrated into Section 5.5 and Table 3 of the revised manuscript.**

---

> ### Author Response · Authors · 2025-11-21
> **Response to Reviewer 9fCa (2/2)**
>
> # For Weakness 5
>
> We utilized a rigorous construction process to ensure the quality of our synthetic data. First, we authored a small set of high-quality seed examples. These were then used to guide LLMs in automated sampling via the prompt provided in Appendix Figure 8. The resulting corpus was subsequently screened against three strict, manual criteria:
>
> 1. Ensuring each generated entry corresponded to a specific, real-world task.
> 2. Guaranteeing the misinformation was a fact-based error highly relevant to the task.
> 3. Verifying coverage across five designated task categories.
>
> **We have incorporated a more detailed explanation of this construction methodology and other key implementation details into the revised manuscript to ensure the dataset's quality and usability are transparent.**
>
> # For Weakness 6
>
> Thank you for your valuable feedback. We acknowledge that the current scale of MisinfoTask is relatively limited. However, a core contribution of our work is providing an easily reproducible dataset construction methodology.
>
> We also recognize that using a single LLM for generation poses a risk of distribution bias. To address this and encourage diversification, we have publicly documented our detailed generation methodology (see Appendix B). We strongly encourage the community to utilize our prompt, adapt it for specific domains, and employ different LLMs and sampling methods to further scale and diversify the testing benchmark.
>
> # For Weakness 7
>
> We confirm that ARGUS primarily targets misinformation related to the core LLM’s internal knowledge, and we acknowledge that defending against dynamic, time-sensitive information requires a more complex, multi-component collaborative defense, a point raised in our Limitations section.
>
> However, we believe the ARGUS methodology is not strictly limited to static knowledge correction. The rectification agent's process extends beyond simple knowledge lookup: it includes identifying potential logical inconsistencies, detecting deviations from common sense, scrutinizing ambiguous phrasing, and executing "goal-aware intent inference". These heuristic checks remain valuable and functional even when facing dynamic scenarios where static knowledge comparison is insufficient. Furthermore, our framework is designed to support the introduction of time-sensitive or dynamic knowledge to both the MAS and the corrective agent via mechanisms like RAG.
>
> # For Question 1
>
> Thank you for your keen observation. We confirm that the value $k=M-1$ mentioned in Appendix B.5 was a critical typo; **the correct configuration is $k=N-1$ (where $N$ is the number of agents).**
>
> In our current setup, selecting $N-1$ edges initially is sufficient to ensure monitoring coverage of all agent nodes. This configuration inherently minimizes the resource cost required for full visibility. **We have already corrected this error in the revised PDF.**
>
> # For Question 2
>
> Thank you for your question. Our CoT strategy is a general, multi-stage rectification framework designed to activate and leverage the LLM's inherent reasoning capabilities and parameterized knowledge. Its universal applicability lies in its structured process, which guides the corrective agent ($a_{cor}$) through methodologically general steps:
>
> 1. Identification. The "Multi-faceted Identification" stage deconstructs the message sentence-by-sentence to identify factual assertions, logical inconsistencies, and ambiguous expressions—a universally applicable analytical step.
> 2. Reasonance. The "Internal Knowledge Reasonance" stage activates relevant internal knowledge clusters of $a_{cor}$ for semantic comparison against incoming information.
> 3. Reconstruction. The "Heuristic Persuasive Reconstruction" stage guides the model to generate corrective explanations that are readily accepted.
>
> The generalizability of this strategy is empirically confirmed across multiple dimensions in our experiments: it consistently reduces Misinformation Toxicity and increases Task Success Rate across 4 different base models, 3 different attack methodologies, and 5 different task types.
>
> ---
>
> We sincerely look forward to further discussion to clarify any other concerns. **If you are satisfied with our reply and the revisions, we kindly hope you will consider increasing our score.**

---

> ### Comment · Reviewer_9fCa · 2025-11-25
>
> Thank you for your response.
>
> For Weakness 1&2, the authors' response regarding the reliance on topological centrality and goal inference is not persuasive. If the  corrective agent fails to infer the correct goal, which is highly probable with subtle misinformation, the primary weight of the detection mechanism fails. The reliance on a weighted sum does not mitigate this.
>
> Regarding Weakness 4, the response regarding the fixed hyperparameters is insufficient. While the authors explain the logic of the need of these, they do not justify the exact static values selected.
>
> For Weakness 6, I disagree with the authors that a such small dataset size can draw concrete conclusion. Drawing broad conclusions about generalization across LLMs based on only 108 samples is statistically unsound. More importantly, since the dataset is generated by LLMs, there is a circular dependency. The evaluation essentially tests if a model can detect misinformation generated by its own family of models. This lacks the diversity and unpredictability of real-world adversarial data.
>
> For Question 2, only experiment scores in this paper cannot verify the generalization ability of this manually defined COT strategy.
>
> Overall, the usage of the proposed method is quite limited as it only focuses on the factual knowledge stored in LLM instead of the widely spread knowledge online. I also find even the only example provided is quite naive as the tools are merely LLM-simulated text outputs, JSON strings returned by a prompt. This leads me to question the quality of the LLM-generated data. Additionally, it assumes, without guarantee, that the generated knowledge is implicitly stored within the model. Given the reasons above, I will keep my rating.

---

> > ### Author Response · Authors · 2025-11-27
> >
> > Dear **Reviewer 9fCa**:
> >
> > We greatly appreciate your valuable feedback and understand your concerns. Below are our further responses to these points.
> >
> > # For Weakness 1&2
> >
> > We acknowledge your concern regarding goal inference accuracy. However, we maintain that our approach is robust for two key reasons:
> >
> > 1. **Resilience via Weighted Aggregation.** By integrating topological, frequency, and relevance scores to select the top-(M-1) edges, the system ensures broad coverage and localization elasticity, preventing failure based on a single metric.
> > 2. **Dynamic Self-Correction.** Goal inference is dynamic, not static. If the agent detects more distinct anomalies in subsequent turns, it updates its context with new inferences. This continuous process dilutes previous errors, allowing the system to adaptively correct its focus and effectively mitigate misinformation propagation.
> >
> > # For Weakness 4
> >
> > The specific values (0.2, 0.2, 0.6) were selected empirically. In our preliminary experiments testing various weight distributions (e.g., uniform weighting), we consistently observed that ARGUS achieved optimal performance when the goal-aware misinformation relevance score was prioritized. This empirical finding aligns with our ablation analysis in Table 3, which confirms that the relevance score contributes most significantly to the defense.
> >
> > Moreover, we acknowledge that while fixed for experimental consistency, these hyperparameters could — and ideally should — be dynamically adjusted in real-world deployments to adapt to varying task scenarios.
> >
> > # For Weakness 6
> >
> > We acknowledge your valid concern regarding the dataset size. We are committed to expanding MisinfoTask in future work, employing more diverse construction methodologies and rigorous human verification to ensure greater diversity and statistical strength.
> >
> > Regarding the concern of circular dependency, we mitigated this risk by evaluating ARGUS on distinct model families beyond the dataset generator (`GPT-4o`), specifically `DeepSeek-V3` and `Gemini-2.0-flash`. The consistent and significant effectiveness of ARGUS across these diverse architectures suggests that our results are not artifacts of circular dependency, thereby validating the utility of both the dataset and the defense framework.
> >
> > # For Question 2
> >
> > We argue that the generalization of our CoT strategy stems from its design as a **fundamental, universal workflow** for information analysis, rather than a task-specific heuristic. The process follows a standard logical sequence: **sentence-level identification** $\rightarrow$ **internal knowledge retrieval for comparative reasoning** $\rightarrow$ **targeted rectification**. As this mirrors a general cognitive approach to error correction, we believe its adaptability across diverse contexts is inherent.
> >
> > ---
> >
> > We thank you again for your time and effort in reviewing our paper. Your comments are very insightful and crucial for revising and improving it. We will continue to optimize our work based on your valuable feedback.

---

### Official Review · Reviewer_A4Ps · 2025-10-22

**Soundness:** 3
**Presentation:** 3
**Contribution:** 3
**Rating:** 6
**Confidence:** 5

**Summary:**

This paper introduced MISINFOTASK, a novel dataset  featuring complex, realistic tasks designed to evaluate MAS robustness against such threats. In addtion, they proposed ARGUS, a two-stage, training-free defense framework leveraging goal-aware reasoning for precise misinformation rectification within information flows. Experiment results show that ARGUS exhibits significant efficacy across various injection attacks for misinformation alleviation and task success rate improvement.

**Strengths:**

1. The paper introduces a novel and practical dataset that enables rigorous evaluation of misinformation robustness in multi-agent systems.
2. The proposed ARGUS framework effectively mitigates misinformation without additional training and improves task completion.
3. The work addresses an important and timely problem of misinformation security that has been largely overlooked in prior MAS research.
4. The paper is clearly written, well organized, and supported by comprehensive experimental validation.

**Weaknesses:**

1. It is recommended that the authors revise the structure of Table 1 so that the model names appear in the first column and the defense types in the second column, making the table layout clearer and improving comparability across models.

2. The paper would benefit from a clearer specification of the threat model, detailing attacker goals, capabilities, and assumptions, which would help strengthen the discussion on the security significance of misinformation propagation in MAS.

3. In MAS, various intelligent agents have different motivations, or mindsets. They have competitive, compromising, and accommodating personalities to achieve their goals. Therefore, how does this information and misinformation spread? In addition, how can the spread of illusions and misinformation be distinguished?

4. The contribution and self-containment of the paper could be improved if the authors provided a more detailed description of the structure and content of MISINFOTASK. In particular, including concrete examples or a summary of task types and misinformation patterns would help readers better understand the dataset’s design and relevance.

5. In the experimental analysis section, it is recommended that the authors add a discussion on the MAS topology

6. How does ARGUS resist the attack of Misinformation in existing MAS?

7. it is recommended that the author reflect the results and discussion of the Appendix in the main text.

**Questions:**

see above

---

> ### Author Response · Authors · 2025-11-21
> **Response to Reviewer A4Ps (1/2)**
>
> Dear **Reviewer A4Ps**:
>
> We sincerely thank you for your thoughtful feedback and valuable suggestions on our work!
>
> We are pleased to clarify and answer the questions and provide some additional experimental results.
>
> # For Weakness 1
>
> Thank you for your suggestion. **We have already modified the structure of Table 1 exactly as recommended.**
>
> # For Weakness 2
>
> Thank you for your suggestion. Our Threat Model, including the attacker's objectives and methodologies, was previously detailed in Appendix E.1.
>
> To enhance clarity and strengthen the safety discussion, **we have expanded it to include an explicit description of the attacker's capabilities and permissions, and have relocated the Threat Model to Section 3.3.**
>
> # For Weakness 3
>
> Thank you for your question. We agree that agent motivations are a critical factor influencing information propagation. Our work primarily focuses on collaborative MAS executing complex tasks, where information spread is fundamentally governed by task dependency and communication topology. We think that compromising or accommodating agents may be more susceptible to misinformation.
>
> Regarding the distinction between illusions (hallucinations) and misinformation, we view this as a matter of internal versus external origins. We classify hallucination as an internal risk: factually incorrect content spontaneously generated by the agent. Conversely, misinformation is an external cause: content that is deliberately constructed or naturally generated to mislead the MAS. Therefore, misinformation propagation is an exogenous process intended to induce endogenous hallucinations in agents, which subsequently leads to erroneous decisions and task failure.
>
> # For Weakness 4
>
> Thank you for your suggestion. We have included some illustrative cases of the MAS workflow (see Appendix F, Figure 7) and a dataset example (Appendix F&G, Figure 9) in the appendix. To provide the necessary structural detail, the key fields for each entry in the MISINFOTASK dataset are defined as follows:
>
> - **user_input**: The specific description of the user's task requirement.
> - **misinfo_goal**: Defines the attacker's objective (the target erroneous belief), visible only to the attacker and the compromised agent(s).
> - **misinfo_argument**: The misleading text constructed to achieve the attack goal, injected via methods such as Prompt Injection, RAG Poisoning, and Tool Injection.
>
> Thus, MisinfoTask is designed to evaluate the impact of misinformation on task completion and the performance of various attack and defense strategies.

---

> ### Author Response · Authors · 2025-11-21
> **Response to Reviewer A4Ps (2/2)**
>
> # For Weakness 5
>
> Thank you for your suggestions. Building upon the analysis in Appendix C.1, where we investigated the relationship between various MAS topologies and MT, we have now supplemented this work by introducing additional topological structures for further evaluation:
>
> - Circle: Agents are connected in a closed loop.
> - Star: All peripheral agents are connected exclusively to a single, central agent.
>
> **The results, using Prompt Injection as the attack and DeepSeek-V3 as the core LLM, are already summarized in Figure 6 in our revised paper.**
>
> Our findings indicate that:
>
> 1. The Self-Determined topology, which tailors its structure to the task's division of labor, exhibits strong connectivity and facilitates better task completion.
>
> 2. Simpler topologies, such as the Circle structure, appear more vulnerable to the influence of misinformation.
>
> 3. ARGUS maintains stable and effective performance across all tested topological structures.
>
> # For Weakness 6
>
> The core of ARGUS's defense strategy lies in its non-intrusive, plug-and-play architecture, which allows for seamless integration into existing MASs.
>
> Specifically, the Localization process operates solely by obtaining the MAS's network topology and monitoring information transmitted over the communication links. The Rectification process only requires intercepting and modifying the information flow on these links, without necessitating any structural changes or modifications to the original MAS framework. Therefore, ARGUS can be effectively deployed as a modular, external defense layer.
>
> # For Weakness 7
>
> Thank you for your suggestion. **We agree with your suggestions, and have integrated the discussions, clarifications, and supplementary experimental results currently in the Appendix into the main body of the revised paper.**
>
> ---
>
> We sincerely look forward to further discussion to clarify any other concerns. If you have additional suggestions, we are committed to addressing them and continuously improving our work based on your feedback.

---

### Official Review · Reviewer_Er5U · 2025-10-27

**Soundness:** 4
**Presentation:** 3
**Contribution:** 3
**Rating:** 8
**Confidence:** 4

**Summary:**

This paper studies how LLM-based MAS are vulnerable to hidden misinformation attacks. It introduces MISINFOTASK, a dataset with 108 complex tasks used to test the robustness of MAS. It also presents ARGUS, a defense framework that does not need extra training.

In the first stage, called Adaptive Localization, ARGUS finds the key communication channels. In the second stage, called Goal-aware Persuasive Rectification, it places a corrective agent on these channels. The agent uses chain-of-thought reasoning to break down messages, detect suspicious claims, compare them with its own knowledge, and create persuasive corrections.

Tests show that ARGUS lowers misinformation toxicity by 28.17% on average and raises the TSR by 10.33% compared to systems without defense.

**Strengths:**

-  The design is original. The synthesis of static topological analysis with dynamic, semantic re-localization based on an inferred misinformation goal is a clever and novel approach. Furthermore, the concept of a "persuasive" corrective agent using CoT reasoning is more advanced than simple fact-checking or edge-pruning defenses.

- The MISINFOTASK dataset is a useful contribution.

- Experiments are thorough. This paper tests across multiple LLM families, multiple modern attack vectors and defense baselines. The analysis is detailed.

- The paper is written with clarity.

- Code is attached. Seems well

- This work addresses a challenge to the adoption of MAS.

**Weaknesses:**

- The paper acknowledges the limit of computational overhead & cost but does not quantify the overhead.

- The initial localization step relies on Edge Betweenness Centrality, which is computationally expensive ($O(V \cdot E)$) and does not scale well to large graphs. This would be a bottleneck for MAS with tens or hundreds of agents.

- The paper's definition of misinformation is "content that contradicts the factual knowledge implicitly stored in the parameters of an LLM." Thus "Internal Knowledge Resonance" relies on this. This makes ARGUS vulnerable to misinformation about dynamic, time-sensitive information pre-training data.

- The choice of $k=M-1$ seems to be a brute-force approach to ensure all critical channels are covered, but it maximizes the cost. How performance degrades with a more sparse and realistic $k$.

**Questions:**

1. See weakness

2. How sensitive is the adaptive re-localization mechanism to the weights $\alpha=0.2, \beta=0.2, \gamma=0.6$? Was a sensitivity analysis or sweep performed to arrive at these values?

---

> ### Author Response · Authors · 2025-11-21
> **Response to Reviewer Er5U (1/2)**
>
> Dear **Reviewer Er5U**:
>
> We sincerely thank you for your positive feedback and valuable suggestions on our work!
>
> # For Weakness 1
>
> Thank you for this valid point. We acknowledge the additional latency and resource costs from ARGUS, as noted in our Limitations section.
>
> As a supplementary experiment, we measured the cost of the ARGUS framework and its constituent modules, using GPT-4o-mini as the core LLM.
>
> |                            | **Cost per 10 Instance** |
> | -------------------------- | ------------------------ |
> | Vanilla                    | ~$0.42                   |
> | Attack                     | ~$0.43                   |
> | ARGUS                      | ~$0.54                   |
> | ARGUS w/o Intent Inference | ~$0.45                   |
> | ARGUS w/o Edge Scoring     | ~$0.52                   |
> | G-Safeguard                | ~$0.51                   |
> | Self-Check                 | ~$0.44                   |
>
> # For Weakness 2
>
> We agree that the computational complexity of EBC is a major concern when dealing with large graphs comprising hundreds of nodes. However, the MAS used in our task-matched scenario are strictly limited to fewer than ten agents. At this specific scale, the overhead associated with calculating EBC during the initial localization phase does not constitute a computational bottleneck.
>
> We acknowledge that extending the ARGUS framework to hyper-scale systems (hundreds of agents) would necessitate the adoption of more efficient centrality estimation algorithms or alternative strategies. This important challenge falls outside the scope of the current task-oriented multi-agent collaboration scenario, and we consider it a valuable direction for future work.
>
> # For Weakness 3
>
> We confirm that the definition of "misinformation" and the "Internal Knowledge Resonance" mechanism primarily target content conflicting with the static knowledge stored in LLM parameters, as we discussed in our Limitations section.
>
> However, ARGUS is not strictly limited to static knowledge correction. "Internal Knowledge Resonance" is merely one component of a broader Goal-aware Reasoning process. The rectification procedure also includes identifying potential logical inconsistencies, checking for deviations from common sense, scrutinizing ambiguous phrasing, and performing goal-aware intent inference. If comparison against static knowledge fails (e.g., due to dynamic, time-sensitive information), these other multi-faceted heuristic strategies retain the capacity to identify the misleading nature of the input.

---

> ### Author Response · Authors · 2025-11-21
> **Response to Reviewer Er5U (2/2)**
>
> # For Weakness 4
>
> Thank you for your reminder! We found this was a critical writing error: **the initial value for the monitoring parameter, $k$, should be $N-1$ (where $N$ is the number of agents), not $M-1$.** We have corrected this in our revised paper.
>
> Setting the number of monitored edges $k$ to $N-1$ in the Edge Localization stage is sufficient to ensure coverage of all nodes in the MAS. We consider this an optimal setting as it maximizes resource efficiency while maintaining full system visibility, thereby addressing the reviewer's concern that our approach maximizes cost.
>
> # For Question 2
>
> Thank you for your question. We performed an initial hyperparameter sensitivity analysis to arrive at our chosen, balanced configuration.
>
> To further investigate the impact of these parameters, we conducted targeted ablation studies on the three scores used for misinformation localization.
>
> |                       | MT   | TSR   |
> | --------------------- | ---- | ----- |
> | ARGUS                 | 3.73 | 75.86 |
> | w/o $\alpha$            | 4.14 | 70.37 |
> | w/o $\beta$             | 3.76 | 72.22 |
> | w/o $\gamma$            | 4.59 | 68.52 |
> | w/o $\beta$ and $\gamma$  | 4.34 | 69.44 |
> | w/o $\alpha$ and $\gamma$ | 4.79 | 67.59 |
> | w/o $\alpha$ and $\beta$  | 3.91 | 73.14 |
>
> These experiments demonstrated that the information relevance score provides the largest contribution to overall defense efficacy. Based on this finding, we adopted a higher weight for $Score_{rel}$, while assigning lower and equal weights to the other two scores. This configuration is logically justified as it prioritizes monitoring high-risk (semantically relevant) information flows.
>
> **We have already integrated these results and the corresponding analysis into Section 5.5 of our revised manuscript.** Thank you for your suggestion.
>
> ---
>
> **We sincerely look forward to further discussion to clarify any other concerns.**

---

> ### Comment · Reviewer_Er5U · 2025-11-25
>
> Thanks for your response. I think the revised paper is better and I will keep my rating. Good luck.

---

### Official Review · Reviewer_TzYe · 2025-10-31

**Soundness:** 2
**Presentation:** 2
**Contribution:** 2
**Rating:** 2
**Confidence:** 3

**Summary:**

This paper studies how misinformation affects Multi-Agent Systems. Specifically, the paper introduces MISINFOTASK which evaluates how MAS defends against misinformation injection. It also proposes ARGUS, a training free defense that (i) locates the most critical communication channels where misinformation is likely flowing, then (ii) performs “goal-aware” persuasive rectification using CoT-style reasoning to counter and correct it.

**Strengths:**

1. The paper is studying a meaningful question that how misinformation propagate within MAS after information injection attacks.
2. The paper builds a complete benchmark including the dataset, setup and evaluation, and also propose a method to tackle such problem.

**Weaknesses:**

1. The paper first uses LLM to generate tasks, and then manually filter out tasks. However, it is unclear whether these tasks align with real-world tasks and whether they are diverse enough to be used, since the same prompt is being used to generate tasks over and over.
2. The dataset only contains 108 tasks, which is small, especially the main content is crafted by LLM.
3. The evaluation employs an LLM judge. Although LLM judge could be useful, the paper doesn't have any sanity check of it, for example, compare it with manual scores.
4. While the paper states that one limit of ARGUS is efficiency and cost, there is no such measurement of its cost compared to other methods. This call into question whether such method is practical, that it could be trading massive resource for performance boost.

**Questions:**

The questions below correspond to each point of the weakness.

1. Can you explain how you ensure diversity and real-world utility of these tasks?
2. (Please see weakness)
3. Do you have any analysis of the LLM judge (human agreement)?
4. Can you provide how much additional cost of the proposed method compared to other methods?

---

> ### Author Response · Authors · 2025-11-21
> **Response to Reviewer TzYe (1/1)**
>
> Dear **Reviewer TzYe**:
>
> We sincerely thank you for your thoughtful feedback and valuable suggestions on our work!
>
> We are pleased to clarify and answer the questions and provide some additional experimental results.
>
> # For Weakness 1 & Question 1
>
> Thank you for your concern regarding the real-world alignment and diversity of MisinfoTask dataset. We ensure both points through a multi-stage process:
>
> - Real-world Alignment. During the manual review stage, our primary criterion was to ensure tasks possessed real-world applicability and to filter out incongruous entries.
> - Diversity. To prevent homogeneity from a single prompt, the prompt in Figure 8 explicitly instructs the AI to generate tasks across five distinct cognitive categories. The subsequent manual filtering ensures these categories are appropriately covered, as shown in Table 3.
>
> Therefore, the dataset construction combines prompt-guided generation with manual review to ensure the dataset's diversity and real-world alignment.
>
> # For Weakness 2 & Question 2
>
> Thank you for your valuable feedback. We acknowledge that the current scale of MisinfoTask is relatively limited. However, one of the core contributions of our work is the provision of a transparent and reproducible method for dataset construction.
>
> We have provided the full prompt used for data generation in the Appendix Figure 8. By specifying different task types and utilizing diverse base LLM models, additional data entries can be sampled to expand the scale of the dataset.
>
> # For Weakness 3 & Question 3
>
> Your concern is well-founded. While LLM-as-a-Judge is a widely adopted paradigm, verifying its alignment with human evaluation is crucial, especially given the complexity of the misinformation tasks in our study.
>
> To validate the reliability of our metrics, we conducted a comparative analysis between the LLM judge and human evaluators across the three attack methods. We assessed the consistency for both MT and TSR. The results, presented in the table below, demonstrate a high degree of alignment, thereby validating the effectiveness of our evaluation approach.
>
> | Prompt Injection | MT(LLM) | MT(Human) | TSR(LLM) | TSR(Human) |
> | ---------------- | ------- | --------- | -------- | ---------- |
> | Attack-only      | 4.94    | 4.48      | 67.74    | 58.33     |
> | ARGUS            | 3.73    | 3.68      | 75.86    | 67.59     |
>
> # For Weakness 4 & Question 4
>
> Thank you for raising this question. As noted in Section 7, introducing an external defense module inevitably incurs additional overhead.
>
> To quantify this impact and address your concern regarding "trading massive resources for performance", we conducted a supplementary experiment to measure the cost and latency of ARGUS compared to other baseline defense methods. We utilized GPT-4o-mini as the backbone LLM for these tests. The results, presented in the table below, indicate that while ARGUS does introduce overhead, the trade-off is within an acceptable range given the significant performance improvements it provides.
>
> |                            | Cost per 10 Instance |
> | -------------------------- | ------------------------ |
> | Vanilla                    | ~$0.42                   |
> | Attack                     | ~$0.43                   |
> | ARGUS                      | ~$0.54                   |
> | ARGUS w/o Intent Inference | ~$0.45                   |
> | ARGUS w/o Edge Scoring     | ~$0.52                   |
> | G-Safeguard                | ~$0.51                   |
> | Self-Check                 | ~$0.44                   |
>
> **We have added these experiments to our paper.** Thank you for your valuable suggestions.
>
> ---
>
> We sincerely look forward to further discussion to clarify any other concerns. If you have additional suggestions, we are committed to addressing them and continuously improving our work based on your feedback. **If you are satisfied with our rebuttal and the revisions, we kindly hope you will consider increasing our score.**

---

> > ### Comment · Reviewer_TzYe · 2025-11-23
> > **Reply**
> >
> > Thank you for your replies. I have re-evaluated my judgement and raised the score.

---

> > > ### Author Response · Authors · 2025-11-24
> > >
> > > Dear Reviewer TzYe,
> > >
> > > We sincerely thank you for your taking the time to review our updated work. We are grateful for your recognition and for increasing the rating - it means a lot to us and inspires us to continue improving.
> > >
> > > Should you have any further questions, please do not hesitate to let us know. We would be happy to engage in further discussion.
> > >
> > > Best regards,
> > >
> > > The Authors

---

### Author Response · Authors · 2025-12-01
**Rebuttal Summary for ACs and SACs**

Dear **Area Chairs**, **Senior Area Chairs**, and **Reviewers**,

We are deeply grateful for your efforts concerning the OpenReview leak incident. We understand the demands on your time and deeply appreciate your willingness to review our paper and provide a fair assessment.

To help Area Chairs clearly understand the discussions during the rebuttal, we have summarized the Strengths raised by the reviewers, and the content and outcomes of our discussions with each reviewer regarding the Weaknesses and Questions, and submit this summary for your reference. We have uploaded the revised version of the manuscript, and all changes are highlighted in blue.

**Crucially, we wish to highlight a key fact regarding our submission:** Reviewer **TzYe** raised their score from 2 to 4 on **November 24th** (prior to the leak incident on November 27th). This adjustment was driven entirely by the quality of our rebuttal, the supplementary experiments, and the improvements made to the manuscript—and was unrelated to any identity leakage.

# Strengths

`All reviewers` unanimously agreed that our work addresses a **significant and timely research problem**. The majority of reviewers (`TzYe`, `A4Ps`, `Er5U`) acknowledged our two primary contributions—the **MisinfoTask dataset** and the **ARGUS framework**—and recognized the thoroughness of our experimental validation. Additionally, Reviewers `Er5U` and `A4Ps` specifically commended the clarity of our writing and presentation.

We believe this consensus reflects the completeness and organization of our work, highlighting its substantial contribution to the field of misinformation in MAS. The reviewers' concerns primarily centered on the need for additional experimental support, multi-perspective analysis, and a broader scope of evaluation.

# Weaknesses

1. **Queries regarding the generation and generalizability of the MisinfoTask dataset, from `TzYe (W1, W2, Q1)` and `9fCa (W5, W6, W7)`.**
	We provided detailed responses to these concerns and incorporated the corresponding discussions and clarifications into the revised manuscript. We emphasized our rigorous generation pipeline: Seed Examples -> AI Generation -> Human Screening, and ensured generalizability by categorizing entries into common misinformation types. Furthermore, we have fully open-sourced our code and dataset generation prompts to guarantee transferability and reproducibility. Notably, `Reviewer TzYe` increased their score following our response.

2. **Questions regarding the rationale and generalizability of the ARGUS framework, from `Er5U (W2, W3, W4)` and `9fCa (W1, W2, W3, Q1, Q2)`.**
	We addressed these questions directly, clarifying the rationale and feasibility of the initial localization step within our threat model, and explaining the effectiveness and generalizability of our Goal-aware Reasoning. These explanations are supported by empirical evidence in the paper. Regarding `Er5U (W4)` and `9fCa (Q1)`, we identified that a misunderstanding was caused by a typo, which has since been corrected. `Reviewer Er5U` acknowledged this, stating: "I think the revised paper is better and I will keep my rating(8)."

3. **Requests for additional experiments, from `TzYe (W3, W4, Q3, Q4)`, `Er5U (W1, Q1)`, `A4Ps (W5)`, and `9fCa (W4)`.**
	Reviewers highlighted the need to verify ARGUS's resource consumption, incorporate alternative evaluation metrics, and conduct further ablation studies on hyperparameters and topological structures. We acknowledged this necessity and dedicated significant time during the rebuttal to conduct extensive new experiments covering all points raised by the reviewers. These results have been integrated into the manuscript, making the paper more substantial and complete.

4. **Suggestions on content completeness and presentation, from `Er5U (Q2, Q3)` and `A4Ps (W1, W2, W3, W4, W6, W7)`.**
	Reviewers provided valuable advice regarding the presentation and organization of the paper. We have revised the manuscript accordingly, refining the prose and integrating new evidence to ensure that our contributions are presented clearly, concisely, and are easily verifiable.

---

**We express our sincere gratitude to all reviewers for their constructive feedback. The suggestions and insights provided have been invaluable in improving our paper. We hope that this summary clarifies the questions raised by the reviewers.**

**We sincerely thank ACs and SACs once more for your valuable time and insightful feedback. Your dedication contributes to making ICLR a more open, inclusive, fair, and equitable research community.**

Best regards,

The Authors

---

### Meta-Review · Area_Chair_3goS · 2025-12-22

**Summary:**

This paper investigates the vulnerability of LLM-based Multi-Agent Systems (MASs) to misinformation attacks by introducing MISINFOTASK, a novel dataset with 108 complex tasks designed to assess MAS robustness. Additionally, the authors propose ARGUS, a two-stage training-free defense framework that locates critical communication channels and employs goal-aware reasoning for misinformation rectification. Experimental results reveal that ARGUS significantly reduces misinformation toxicity and enhances task success rates in the presence of such attacks.

**Reviewer Concerns:**

All the weaknesses were mostly addressed in my opinion:

- *Lack of Diversity in Generated Tasks*: Reviewers wished authors had clarified how the tasks generated using LLMs align with real-world scenarios and whether they ensure diversity, especially since the same prompt and LLM is used for generation. **Authors addressed this by pointing out that they carefully curate the LLM outputs to align with real-world applicability and enforce diversity within their prompting framework.**
- *Small Dataset Size*: Two reviewers noted that the dataset containing only 108 tasks is relatively small and may not be sufficient for robust benchmarking, particularly given its reliance on LLM-generated content. **Authors somewhat addressed this concern by specifying that their LLM task generation framework is fully reproducible, but they should also clarify why scaling up is challenging or consider expanding the dataset.**
- *Evaluation Methodology*: Reviewers felt that the evaluation using an LLM judge lacked a sanity check against manual scoring, which is necessary to validate the LLM's assessment abilities. **Authors agreed and addressed this by running a human evaluation to corroborate LLM-judge results. However, authors should also compute inter-annotator agreement metrics to strengthen their claims.**
- *Quantification of Costs*: Reviewers felt the paper should quantify the overhead costs associated with their method to better assess its practicality and efficiency compared to alternative methods. **Authors expanded on this limitation by computing overhead costs more directly, addressing this concern.**
- *Dependence on Specific Metrics*: Reviewers noted that the reliance on Edge Betweenness Centrality in the pipeline may not scale well for larger multiparty systems and may allow adversaries to exploit less central paths. **Authors acknowledged this limitation and pointed out they limit their experiments to a maximum of 10 agents. However, they should better discuss the challenge of scalability in their limitations.**
- *Transparency in Dataset Construction*: Reviewers were concerned about the transparency of the methodology used in the dataset's manual review process, specifically regarding the qualifications of evaluators and clear filtering guidelines. **Authors addressed this by clarifying their methodology and committing to adding more details in the paper.**
- *Challenges of Dynamic Misinformation*: Reviewers expressed that the approach may overlook the complexities of dynamic, time-sensitive misinformation, which could affect the dataset's applicability to real-world scenarios. **Authors acknowledged this limitation while suggesting that their ARGUS method could help address dynamic misinformation, although future work is needed to confirm this.**

**Reviewer Scores:**

One reviewer had increased their score before the leak, I believe the others would have too.

---

### Decision · Program_Chairs · 2026-01-26

Accept (Poster)